# Controlling the Surface Morphology of Two-Dimensional Nano-Materials upon Molecule-Mediated Crystal Growth

**DOI:** 10.3390/nano13162363

**Published:** 2023-08-18

**Authors:** Tetsuo Yamaguchi, Hyoung-Jun Kim, Hee Jung Park, Taeho Kim, Zubair Khalid, Jin Kuen Park, Jae-Min Oh

**Affiliations:** 1Department of Energy and Materials Engineering, Dongguk University, Seoul 04620, Republic of Koreataeho0408@naver.com (T.K.);; 2Plasma Convergence R&BD Division, Cheorwon Plasma Research Institute, 1620, Hoguk-ro, Galmal-eup, Cherwon-gun 24047, Gangwon-do, Republic of Korea; hjun.kim@yonsei.ac.kr; 3KBSI Western Seoul Center, University-Industry Cooperation Building, 150 Bugahyeon-ro, Seodaemun-gu, Seoul 03759, Republic of Korea; hjpark8659@kbsi.re.kr; 4Department of Chemistry, Hankuk University of Foreign Studies, Yongin 17035, Gyeonggi-do, Republic of Korea

**Keywords:** surface, alkaline earth oxides, clays, MgO

## Abstract

The surface morphology of Mg-Al-layered double hydroxide (LDH) was successfully controlled by reconstruction during systematic phase transformation from calcined LDH, which is referred to as layered double oxide (LDO). The LDH reconstructed its original phase by the hydration of LDO with expanded basal spacing when reacted with water, including carbonate or methyl orange molecules. During the reaction, the degree of crystal growth along the ab-plane and stacking along the c-axis was significantly influenced by the molecular size and the reaction conditions. The lower concentration of carbonate gave smaller particles on the surface of larger LDO (2000 nm), while the higher concentration induced a sand-rose structure. The reconstruction of smaller-sized LDH (350 nm) did not depend on the concentration of carbonate due to effective adsorption, and it gave a sand-rose structure and exfoliated the LDH layers. The higher the concentration of methyl orange and the longer the reaction time applied, the rougher the surface was obtained with a certain threshold point of the methyl orange concentration. The surface roughness generally increased with the loading mount of methyl orange. However, the degree of the surface roughness even increased after the methyl orange loading reached equilibrium. The result suggested that the surface roughening was mediated by not only the incorporation of guest molecules into the LDH but also a crystal arrangement after a sufficient amount of methyl orange was accommodated.

## 1. Introduction

Controlling surface morphology such as roughness has attracted particular attention in bulk materials in terms of energy saving and environmentally friendly applications. Surface controlled materials are widely utilized as antireflection surfaces [1,2,3,4], light harvesting films [5,6,7], superhydrophobic surfaces [8,9,10,11,12], etc. In parallel, the surface control of an individual particle composing bulk material on the nanoscale is also an important issue considering the fabrication and engineering for versatile applications, for example, heterogeneous catalysts [13,14], biological cellular uptake [15,16], and so on. Layered double hydroxide (LDH), one of the layered materials with a chemical formula of [M^2+^_1−x_M^3+^_x_(OH)_2_]^x+^(A^n−^)_x/n_∙mH_2_O (M^2+^: divalent metal, M^3+^: trivalent metal, A^n−^: interlayer anion, m: number of interlayer water), is a biologically and environmentally friendly material [17,18,19,20] and is easily synthesized under mild conditions [21,22] to have high anisotropy [23,24]. LDH is a well-known precursor for mixed metal oxide [25,26], of which the particle morphology is fairly dependent on that of the precursor, LDH [27,28,29,30]. From a macroscopic point of view, LDH particles tend to form a rough surface resulting from interparticle edge-to-face interactions by forming a house-of-cards structure, and this was exploited as an antireflection film [31] and photocontrol of the water contact angle [32]. Controlling the surface roughness of an individual LDH particle is an attractive topic in terms of microscopic surface modification, such as the effective adsorption of dye [33,34] or gas molecules [35,36]; however, there has been only limited research on this issue.

We expected, in this study, that the surface roughness of an LDH particle could be controlled if the reconstruction condition from a mixed metal oxide, also called layered double oxide (LDO), to LDH is regulated. It is empirically known that LDOs recover the original crystal structure through simultaneous hydration and intercalation of anionic species [37,38,39]. Metal hydroxides formed resulting from hydrolysis of metal oxide would undergo two crystal growth processes during the reconstruction of the LDH structure: (i) metal hydroxide layer propagation along the ab-plane and (ii) stacking along the c-axis. The competition between those two processes could result in various types of surface roughness in an individual LDH particle. Although general physicochemical changes of LDHs during reconstruction, such as crystal phase, global crystallinity, porosity, etc., are fairly well-known [39,40,41,42,43], detailed research on the surface roughness through competitive crystal growth is quite limited so far [44,45,46].

In this report, we are going to demonstrate how the surface roughness of the individual LDH particle is controlled through reconstruction with the presence of small CO_3_^2−^ and large organic methyl orange (MO) anions, which can be incorporated with LDHs by various supramolecular interactions, such as ionic, van der Waals, and π-π interactions [47,48,49,50]. With the goal of establishing reaction parameters, such as time and concentration, the surface roughness upon changing lattice morphologies and sizes was systematically monitored. Therefore, the obvious relationship between the surface roughness of the particles and the loading amount of the anions was investigated. Furthermore, the roughness of the particles was profound as the reaction time elapsed longer. Interestingly, it was found for the first time that there was a certain threshold amount of organic substance to mediate the surface morphology, as the detailed mechanism illustrated in Figure 1 shows. The rougher surface was obtained at a higher loading of MO and with a longer reaction time.

## 2. Materials and Methods

### 2.1. Materials

Aluminum nitrate nonahydrate (Al(NO_3_)_3_·9H_2_O), magnesium nitrate hexahydrate (Mg(NO_3_)_2_·6H_2_O), urea (CH_4_N_2_O), sodium bicarbonate (NaHCO_3_), and methyl orange (C_14_H_14_N_3_NaO_3_S) were purchased from Sigma-Aldrich Co., LLC. (St. Louis, MO, USA). Sodium hydroxide (NaOH) in pellet form was purchased from Daejung Chemicals & Metals Co., Ltd. (Siheung, Republic of Korea). All reagents were used without further purification.

### 2.2. Synthesis

#### 2.2.1. MgAl-LDH with a Diameter of 2000 nm (LDH_2000_) [44,51]

The LDH_2000_ was obtained by the urea hydrolysis method. A mixed metal solution was prepared by mixing 0.66 M Mg(NO_3_)_2_·6H_2_O and 0.33 M Al(NO_3_)_3_·9H_2_O aqueous solutions. Solid urea was added to the mixed metal solution until the molar ratio of urea/(Mg^2+^ + Al^3+^) reached a value of 3.3. The transparent solution was heated at 90 °C, under stirring for 2 days, to obtain a suspension. The obtained suspension was transported to a Teflon®-lined stainless-steel bomb and then aged at 100 °C for 2 days. The resulting precipitate was lyophilized after centrifugation and washed with deionized water.

#### 2.2.2. MgAl-LDH with a Diameter of 350 nm (LDH_350_)

The LDH_350_ was synthesized by coprecipitation followed by hydrothermal treatment, as reported previously [52]. A mixed metal solution of 0.3 M Mg(NO_3_)_2_·6H_2_O and 0.15 M Al(NO_3_)_3_·9H_2_O was titrated with an alkaline solution containing 0.9 M NaOH and 0.7 M NaHCO_3_ until the pH reached 9.5. The suspension was placed in a Teflon®-lined stainless-steel bomb and aged at 150 °C for 3 days. White precipitates were collected by centrifugation, thoroughly washed with deionized water, and then lyophilized.

#### 2.2.3. Calcination of LDH_2000_ and LDH_350_ (LDO_2000_ and LDO_350_)

The layered double oxides (LDO_2000_ and LDO_350_) were obtained by calcination of LDH_2000_ and LDH_350_ in a muffle furnace at 400 °C for 9 h, respectively.

#### 2.2.4. Reconstruction of the LDHs with CO_3_^2−^

NaHCO_3_ was used in the reconstruction of LDO_2000_ and LDO_350_ as a carbonate source, and the amount of Na^+^ concentration was set the same as that for MO. LDO_2000_ and LDO_350_ (50 mg for each) were added to aqueous NaHCO_3_ solutions (50 mL) at concentrations of 1.52 × 10^−3^ and 2.3 × 10^−2^ M (corresponding to 0.2 and 3 equivalents of CO_3_^2−^ to Al^3+^). The suspensions were stirred for 0.5, 4, and 12 h, respectively. The precipitates were centrifuged and washed with deionized water 4 times. 

#### 2.2.5. Reconstruction of the LDHs with Various MO Concentrations and Reaction Times

The LDOs (50 mg) were added to aqueous methyl orange (MO) solutions (50 mL) at concentrations of 1.52 × 10^−3^, 3.8 × 10^−3^, 7.6 × 10^−3^, and 2.3 × 10^−2^ M (corresponding to 0.2, 0.5, 1, and 3 equivalents of MO to Al^3+^), and the suspensions were stirred for 0.5, 4, and 12 h. The resulting precipitates were centrifuged and washed with deionized water 4 times. The pH values of the suspensions did not significantly change by the reaction time, as summarized in Appendix A. 

### 2.3. Characterization

The crystal structure of all the samples was investigated by a powder X-ray diffractometer (PXRD: D2 Phaser, Bruker AXS GmbH, Karlsruhe, Germany) with Ni-filtered Cu-Kα X-ray (λ = 1.5406 Å). X-ray diffraction patterns were obtained in the 2θ range, 2–30°, with time step increments of 0.02° and 0.5 s/step.

The particle size and morphology were visualized by using field emission scanning electron microscopy (FE-SEM; QUANTA 250 FEG, FEI, Hillsboro, OR, USA, and SU8220, Hitachi, Tokyo, Japan at KBSI Western Seoul Center). For preparing samples for SEM measurement, the sample suspensions were put on a Si wafer and then dried in vacuo. The samples immobilized on the Si wafer were attached on a stub using carbon tape and sputtered with Pt/Pd plasma with 20 mA for 60 s. The sample images were obtained under 30 kV acceleration voltage.

Surface roughness and height profiles of all samples were obtained using atomic force microscopy (AFM; NX-10, Parksystems, Suwon, Republic of Korea). In order to measure AFM, sample suspensions were dropped on the Si wafer and dried under ambient conditions. The sample images and height profiles were obtained by non-contact mode with 0.5 Hz scan speed. The AFM images were analyzed using XEI 1.6 software provided by Parksystems. 

In order to calculate the loading ratio of MO, the concentrations of MO in the supernatants before and after the reaction, as well as in the washing supernatant, were quantified by using a UV-vis spectrometer (UV-1800, SHIMADZU, Kyoto, Japan). The loading ratio, *r%*, of MO was estimated from Equation (1). The *MO_obs_* stands for the experimentally absorbed amount (g) of MO uptake during LDO’s phase transformation, and *MO_calc_* indicates the theoretically calculated amount (g) of MO, which can be taken up by the charge neutralization between LDH and MO. *MO_obs_* was calculated from the absorbance at 470 nm of MO that remained in the supernatant and in aqueous solutions used for washing; *MO_calc_* was calculated with hypothesizing that MO was fully intercalated into LDH through charge neutralization.
(1)$$r\%=\frac{{MO}_{obs}}{{MO}_{calc}}\times 100$$

## 3. Results

### 3.1. XRD Patterns of the LDHs and the LDOs

Powder X-ray diffraction (XRD) patterns of the LDHs and the LDOs are shown in Figure 2. Both LDH_2000_ and LDH_350_ exhibited similar diffraction patterns with peaks at 11.7° and 23.5°, which were attributed to the reflection of (003) and (006), as shown in Figure 2(a,b). The diffraction peak at 11.7° indicated that the basal spacings of both the LDHs were 0.75 nm, corresponding to the carbonate interlayered LDH. There are also clearly observed characteristic diffraction peaks of (012), (015), (018), (110), and (113) at 2θ values of 35°, 40°, 47°, 61°, and 62°, suggesting the sufficiently high crystallinity of both the samples [53,54,55].

Upon heat treatment at 400 °C for 9 h, all the diffraction peaks of the LDHs disappeared, as shown in Figure 2(c,d). There were observed (111), (200), and (220) peaks at 36°, 45°, and 62°, which are typical patterns for rock-salt type MgO [44,55,56]. It is well-known that MgO nanocrystallites are developed during mild heat treatment and that those crystallites are interconnected by tetrahedral Al^3+^, which is migrated from the octahedral site of LDH [45]. Thus, we could not find any crystalline signals of AlO_x_ moiety in the XRD patterns of the calcined LDHs [57].

### 3.2. SEM Images of the LDHs and the LDOs

As shown in the SEM images (Figure 3a,c), both LDH_2000_ and LDH_350_ had hexagonal shapes with smooth surfaces. As shown in Appendix A, the surface roughness, R_a_, was estimated to be 37 nm and 0.8 nm, respectively, for LDH_2000_ and LDH_350_ from the AFM. As the R_a_ and root-mean-square (RMS) values of single LDH particles were less reliable due to a tilted arrangement of the LDH_2000_ and unclear particle edge of the small particle size of the LDH_350_, we considered these values as a tendency of the surface roughness rather than the quantitative results. The reason why LDH_2000_ had a rougher surface than LDH_350_ is due to the large lateral dimension. The SEM images of LDO_2000_ and LDO_350_ in Figure 3b,d showed that the hexagonal shapes were maintained after the heat treatment. The surface flatness was also maintained after the heat treatment, according to the surface roughness of 35 nm and 0.8 nm, respectively, for the LDO_2000_ and LDO_350_ (Appendix A). The change of morphology due to the conversion from LDH to LDO was negligibly small in terms of the microscopic observation. An LDH crystal has an edge-shared M(OH)_6_ octahedron along the ab-plane, and thus its surface can be very flat. The LDO contains Al^3+^(tetrahedron)-mediated MgO domains through an interconnected manner on the surface, and, therefore, it might have a rougher surface. However, the MgO domains are very small in dimension, and their array is fairly homogeneous, resulting in the flatness on the surface. Similar morphology preservation of LDHs after calcination was reported in the previous literature [44,46,58].

### 3.3. XRD Pattern of the Reconstructed LDHs

Figure 4a,b represent powder XRD patterns of LDH_2000_ reconstructed with CO_3_^2−^ at 0.2 eq and 3 eq to Al^3+^ at several reaction times. In both reaction systems, the diffraction peaks attributable to LDO_2000_ decreased, and those of LDH_2000_ increased by reacting with CO_3_^2−^, which indicated the LDH phase was reconstructed by the hydration of the LDO phase with the intercalation of CO_3_^2−^. At 4 h of the reconstruction, the clearer (003) and (006) peaks attributable to the LDH were observed in the 3 eq-treated sample than that reconstructed in 0.2 eq, while the XRD patterns at 12 h were similar in both concentrations. As summarized in Appendix A, the crystalline size of the reconstructed LDH after 4 h at 3 eq, which was estimated from Sherrer’s equation (19.3 nm) and was larger than that reconstructed at 0.2 eq for 4 h (4.66 nm); however, that after 12 h at 3 eq (9.01 nm) was similar to that at 0.2 eq (8.63 nm). It was thought that the reconstruction of LDH at high and low concentrations of CO_3_^2−^ gave different nanostructures.

The clearer growth of the peaks attributed to the LDH and the decrease of the peaks of the LDO were shown in the LDH_350_ systems compared to the LDH_2000_ systems (Figure 4c,d). The higher surface area that originated with the smaller particle size of the LDH_350_ occurred as a result of effective adsorption and reconstruction of the LDH layers. As summarized in Appendix A, the particle sizes of the reconstructed LDH for the 0.5 h reaction estimated by Sherrer’s equation were 8.02 nm at 0.2 eq and 9.03 nm at 3 eq, which were comparable to the ones obtained in the LDH_2000_ system for the 12 h reactions. It suggested that the quick hydration along the ab-plane reconstructed the larger area simultaneously to give the larger LDH particles.

Figure 5 and Figure 6 represent low-angle powder XRD patterns of the LDH_2000_ and LDH_350_ before and after reconstruction with MO at various concentrations and reaction times. As can be seen, the (003) peaks of the pristine LDHs shifted from 2θ = 11.7° to 3.5°, indicating the intercalation of the MO between the LDH layers. The (003) peak intensity of 0.5–3 eq was higher than that of 0.2 eq as the higher MO concentration, leading to effective intercalation and interlayer molecular arrangement of the MO [49]. We could also find that the crystallinity along the (003) plane generally became better upon the reaction time. These results suggest that the interlayer molecular arrangement of MO became ordered upon the reaction time. The peak around 2θ = 5°, which is only seen with 3 eq, might be attributed to the inter-molecular packing among the MO moieties.

### 3.4. SEM Images of the Reconstructed LDHs

Figure 7 shows the SEM images of the reconstructed LDH_2000_ in NaHCO_3_ solutions at 0.2 eq and 3.0 eq to Al^3+^ in the LDO_2000_ at several reaction times. At the low concentration of CO_3_^2−^ (0.2 eq), the surface of the reconstructed LDH_2000_ was covered by small particles. The number of surface particles seemed to be larger at the longer reaction time (Figure 1e). According to the increase of the LDH phase in the XRD patterns (Figure 4) and the size of the crystal (Appendix A), the surface particles were thought to be reconstructed LDH. The relatively small-sized CO_3_^2−^ is adsorbed on the surface of the LDO_2000_ and then incorporated through the surface defect with the reconstruction of the LDH layer to give the rough surface. The reconstruction at 3 eq for 0.5 h did not change the surface morphology significantly, while particles with a sand-rose structure were observed at 4 h. The rough surface with small particles on the sand-rose structure appeared by further reaction for 8 h (a total time of 12 h). It is thought that the adsorption of the CO_3_^2−^ was slow at the low concentration of 0.2 eq so that the surface along the ab-plane was hydrated to give small particles of metal hydroxide about 5 nm. On the other hand, the adsorption of the CO_3_^2−^ was fast at a high concentration of 3 eq to develop CO_3_^2−^ intercalated particles and collapse the larger ab-plane, the formation of relatively large particles of about 20 nm, resulting in the sand-rose structure at 4 h. The massive consumption of CO_3_^2−^ before the 4 h reaction induced the rough surface, generating small particles on the sand-rose surface. The phenomenon corresponded to the decrease of the particle size estimated from Sherrer’s equation from 19.3 nm to 9.01 nm, as summarized in Appendix A.

The reconstruction for 0.5 h with 0.2 eq and 3 eq of CO_3_^2−^ did not alter the morphology of LDO_350_ significantly (Figure 7 right columns). The reconstruction at 0.2 eq for 4 h exfoliated a part of the LDH layers and the sand-rose-like structure, and thus thinner LDH particles were obtained by the 12 h reaction. A similar change of the morphology was observed by the reaction at 3 eq for 4 h. The 12 h reaction time gave a clear sand-rose structure and thinner particles. It is worth noting that the small particles, which were observed in the reconstruction of the LDH_2000_, were not observed in the LDH_350_ systems, suggesting the size effect on the reconstruction by CO_3_^2−^. The larger lateral size of the LDH_2000_ would not allow the exfoliation of a whole layer by the reconstruction, while the whole layer of LDH_350_ was exfoliated, giving the thinner particles. 

Figure 8 shows the SEM images of LDH_2000_ reconstructed under four different MO concentrations (0.2 eq, 0.5 eq, 1 eq, and 3 eq to Al^3+^ in LDO_2000_) at 0.5 h, 4 h, and 12 h. In contrast to the CO_3_^2−^ systems, the smooth surface of the LDO_2000_ was maintained at the concentration of 0.2 eq, with a 16–21 *r*%. The slower reconstruction with MO than that with CO_3_^2−^, which has a strong affinity with LDH, would be a reason. At the concentration of 0.5 eq, the smooth surface was also preserved at 4 h with 41 *r*%, while a rough surface began to appear after 12 h with 44 *r*%. In the case of the 1 eq concentration, the loading ratio increased up to 48 *r*% in the 0.5 h, and surface roughening began to appear. According to the relationship between the surface roughness and the loading ratio shown in Figure 8, it can be suggested that the initial smooth surface was maintained up to 41 *r*% of the MO loading, while an obvious rough surface began to be seen at over 44 *r*%; in other words, a threshold loading ratio to roughen the surface was ~45 *r*% (Figure 1c,d). It is thought that the rigid LDO structure, which is structured by covalent bonds, did not accept the increase of the c-axis (the interlayer space) induced by the crystal growth of the LDH along the ab-plane at less than 45 *r*% (Figure 1d), while the larger *r*% than the threshold collapsed the LDO structure to give a rough surface (Figure 1g). It should be noted that the 1 eq-treated sample showed severe surface roughness change upon the reaction time despite the comparable loading ratio (48 and 46 *r*%), suggesting that the surface roughness progressed even after the MO uptake was equilibrated. Taking into account the XRD results shown in Figure 5c, a probable reason for the surface roughening at the equilibrium is the molecular rearrangement of MO in the interlayer space of the LDH to reach a thermodynamically stable state by π-π interactions [49]. It is worth noting here that the small particles, which were observed in the CO_3_^2−^ systems, were not observed, even at a high loading ratio (~100 *r*%), which suggests that the size of intercalated molecules affected the morphology of the reconstructed LDHs. The 3-dimensional AFM images and height profiles (Appendix A) also showed that both the loading ratio and reaction time affected the surface roughness, R_a_, of the LDH_2000_, although they were rather qualitative compared to the SEM images. The R_a_ of the LDH_2000_ reacted at the 0.2 eq for 0.5 h was 39 nm, which was comparable with pristine LDH_2000_ (R_a_ ~ 37 nm), while the 1 eq at 12 h-treated sample had a significantly enhanced R_a_ of 105 nm. The reconstruction of the LDHs was reported as a memory effect by mixing the LDOs with water and guest anions [40,41,43,46,59], where the small MgO crystallites grew to the LDH layers with ordering at the surface. However, large MO molecules would prohibit an ordered arrangement of the crystallites at the surface through the expansion of the gallery space (Figure 1g) [37,44]. 

The LDH_350_ samples reconstructed with MO showed a similar trend compared with the LDH_2000_, as shown in Figure 9. The smooth surface of the LDO_350_ was fairly well-preserved until 12 h at 0.2 eq, showing loading ratios of 15–19 *r*%. On the other hand, we could observe a rough surface at 12 h of the reaction time when 0.5 eq was applied. In the case of the 0.5 eq-treated sample, the surface roughness was different at 4 h and 12 h, in spite of the comparable loading ratio of 45 *r*%. As can be figured out with the results of the LDH_2000_, the surface roughening seemed to progress after the MO incorporation was equilibrated. When the MO concentrations of 1 eq and 3 eq were applied, all the reconstructed LDHs showed a discrete surface roughness at all the time points with loading amounts higher than 45 *r*%. This suggested that the threshold loading ratio was also 45 *r*% for LDO_350_. Moreover, the R_a_ of the LDH_350_ was governed by both *r*% values of the MO and reaction time, as shown in the AFM images in Appendix A (R_a_ = 0.9 nm at 0.2 eq for 0.5 h and 3 nm at 1 eq for 12 h), although the changes were not sufficiently clear due to the small particle size.

## 4. Conclusions

To summarize, the surface roughness of MgAl-LDH was successfully controlled by taking advantage of the reconstruction of LDO in the presence of small and large guests. The surface roughness was attributed to the denaturation of layer stacking by the intercalation of small CO_3_^2−^ and large methyl orange molecules into the LDH during the reconstruction of the LDO. The change in the surface morphology of the LDH by the reconstruction was controlled by the concentration of the small-sized guest (CO_3_^2−^) and the loading ratio of the large-sized guest (methyl orange, MO), respectively. An obvious size effect of the LDO precursors on the resulting morphology after reconstruction by CO_3_^2−^ was observed. The reconstruction of larger-sized LDH (2000 nm) at low concentrations of CO_3_^2−^ (0.2 eq) caused the small LDH particles on the external surface to give a rough surface (Figure 1e), and a sand-rose morphology was given at the high concentration of CO_3_^2−^ (3 eq, Figure 1f), suggesting the possibility that the reaction rate affected the resulting morphology. The reconstruction of the LDH with a smaller lateral size (350 nm) by the CO_3_^2−^ partially exfoliated the LDH layers rather than giving small particles at the surface (Figure 1f). The partial exfoliation of the layers was only facilitated in the smaller lateral size in the whole layer region. The surface roughening by the intercalation of the MO had a threshold of incorporation (~45%) regardless of the particle size of the pristine LDHs (Figure 1g). The higher incorporation and the longer reaction time gave the rougher surface. As shown in the time-dependent loading ratio values, the MO uptake was equilibrated in an early time, while the roughening continued afterwards. This strongly suggested that the adsorption and the roughening occurred in a different time stage. The CO_3_^2−^, which has a smaller and more rigid structure and higher affinity to the LDH than the MO, effectively adsorbed and collapsed the LDO structure to alter the morphology even at low concentrations and in a short reaction time; on the other hand, MO, which has a more flexible structure than CO_3_^2−^, could maintain the crystalline structure of the LDO until 45% of the leading ratio. It is, therefore, concluded that by choosing appropriate reaction conditions for reconstruction, such as the type of guest molecules, concentration of anions, and reaction time, the surface roughness of an individual LDH particle could be controlled. Furthermore, the control of the roughness of the particle surface could expand the application of LDHs and LDOs from versatile nanomaterials to practical applications, like functional film components.

## Figures and Tables

**Figure 1 nanomaterials-13-02363-f001:**
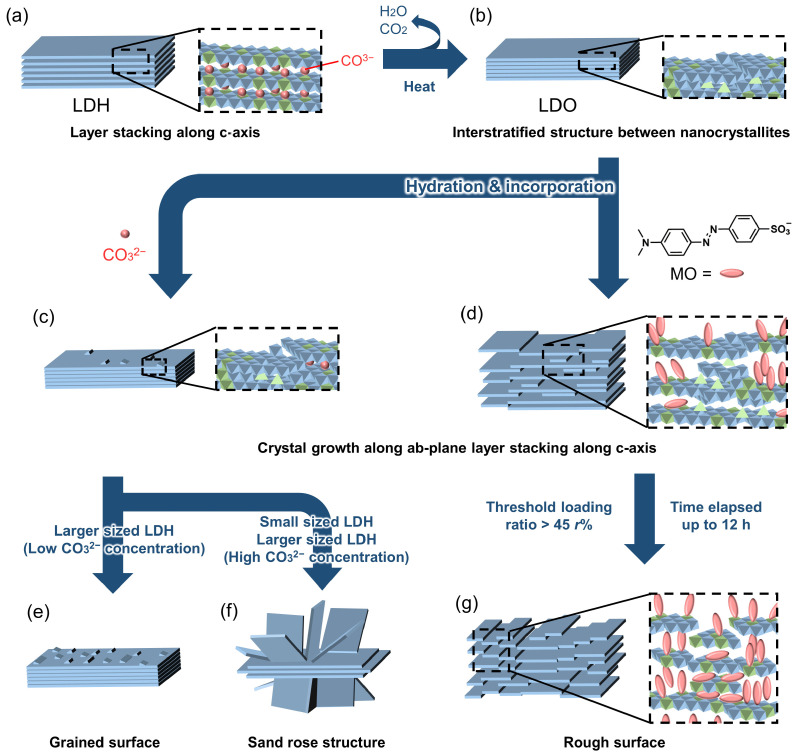
Schematic mechanism of surface roughening of LDH by systematic reconstruction with CO_3_^2−^ and MO. (**a**) pristine LDH, (**b**) LDO obtained by calcination of LDH, and reconstructed LDHs for short time by CO_3_^2−^ (**c**) and MO (**d**) and at the final stages by CO_3_^2−^ (**e**,**f**) and MO (**g**).

**Figure 2 nanomaterials-13-02363-f002:**
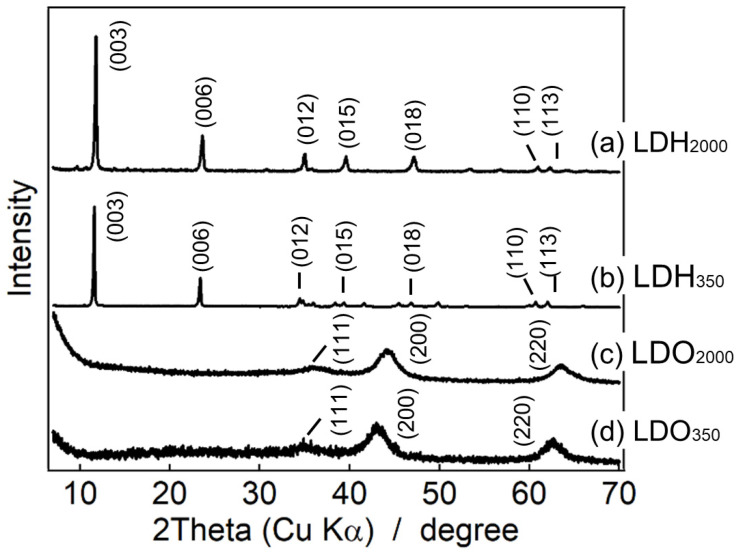
Powder XRD patterns of (**a**) LDH_2000_, (**b**) LDH_350_ divided 10, (**c**) LDO_2000_, and (**d**) LDO_350_ multiplied 5, respectively.

**Figure 3 nanomaterials-13-02363-f003:**
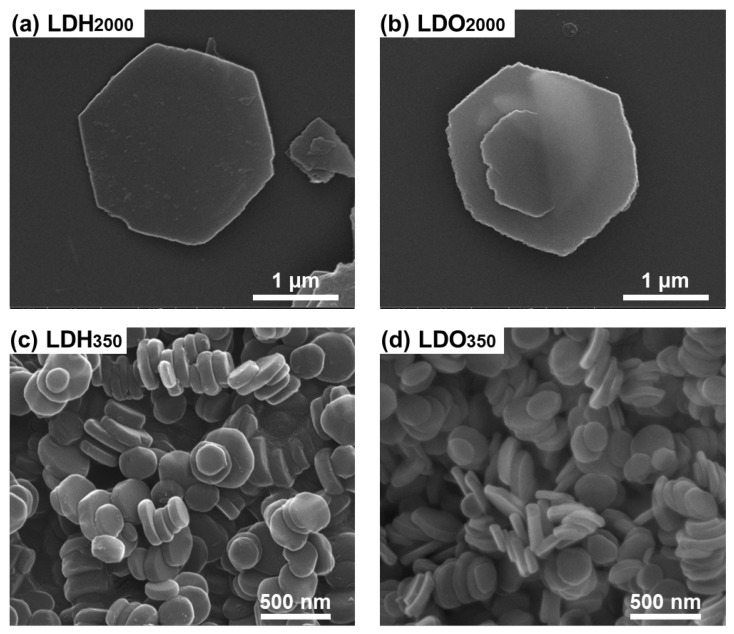
Scanning electron microscopic (SEM) images of (**a**) LDH_2000_, (**b**) LDO_2000_, (**c**) LDH_350_, and (**d**) LDO_350_, respectively.

**Figure 4 nanomaterials-13-02363-f004:**
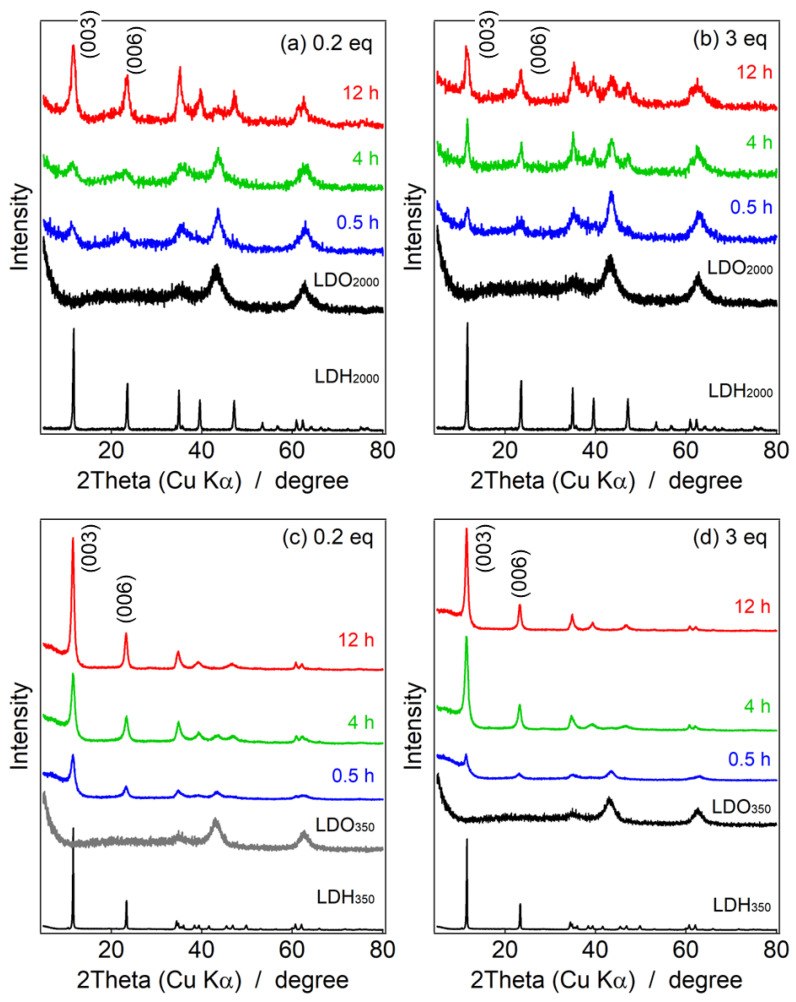
Powder XRD patterns of (**a**,**b**) reconstructed LDH_2000_ and (**c**,**d**) LDH_350_, where bottom black lines: pristine LDHs (LDH_2000_ and LDH_350_), top black lines: LDO precursors (LDO_2000_ and LDO_350_) and reconstructed LDHs with CO_3_^2−^ at the reaction time of blue lines: 0.5 h, green lines: 4 h and red lines: 12 h at the concentrations of CO_3_^2−^ of (**a**,**c**) 0.2 eq and (**b**,**d**) 3 eq.

**Figure 5 nanomaterials-13-02363-f005:**
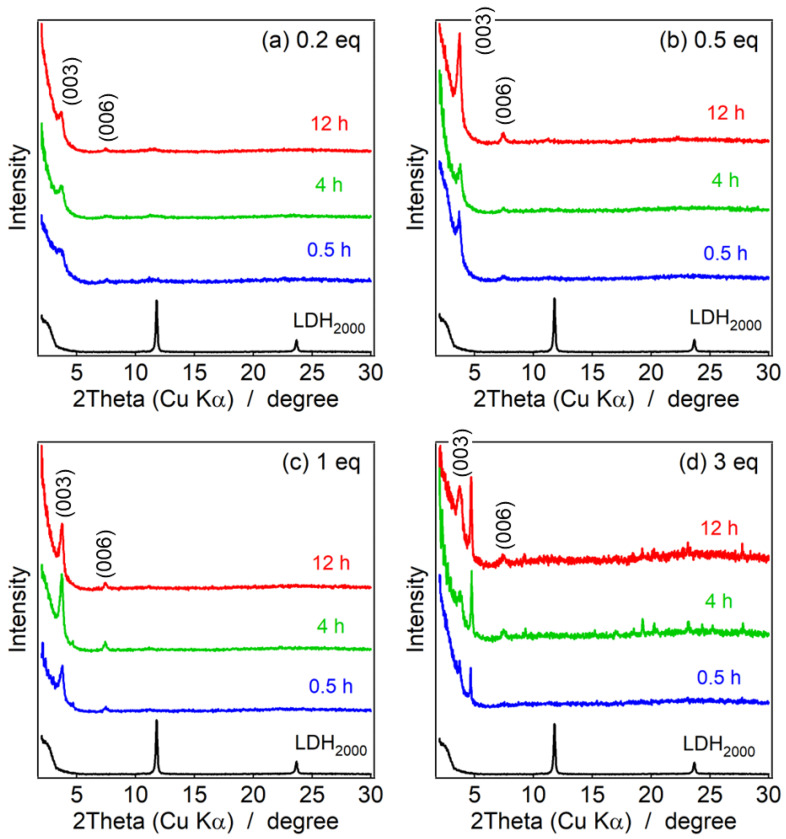
Powder XRD patterns of (black line) LDH_2000_ and reconstructed LDHs at the reaction time of 0.5 h (blue line), 4 h (green line), and 12 h (red line) at the concentrations of MO of (**a**) 0.2 eq, (**b**) 0.5 eq, (**c**) 1 eq, and (**d**) 3 eq.

**Figure 6 nanomaterials-13-02363-f006:**
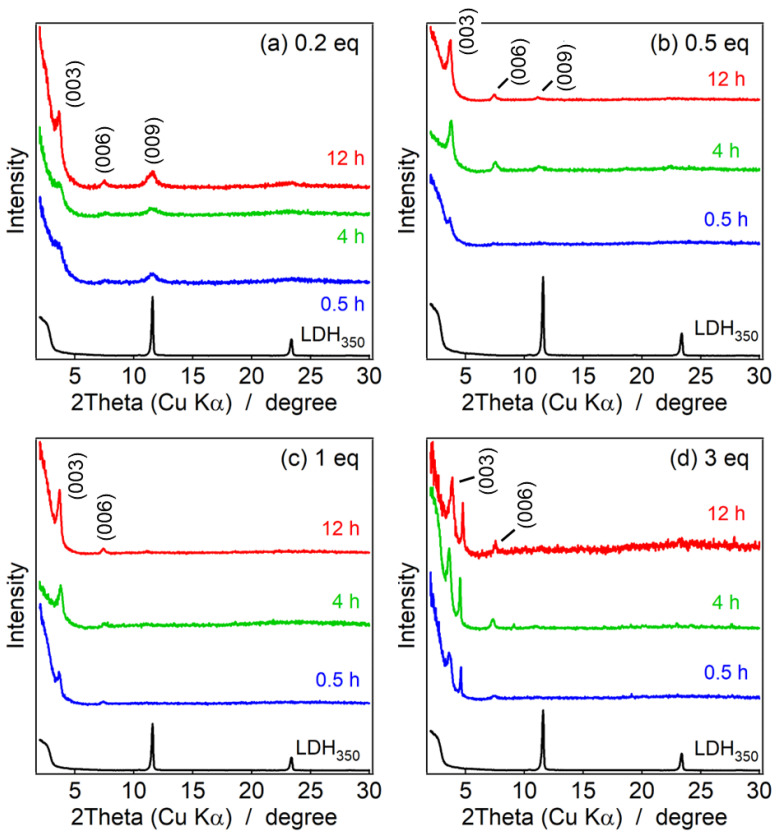
Powder XRD patterns of (black line) LDH_350_ and reconstructed LDHs at the reaction time of 0.5 h (blue line), 4 h (green line), and 12 h (red line) at the concentrations of MO of (**a**) 0.2 eq, (**b**) 0.5 eq, (**c**) 1 eq, and (**d**) 3 eq.

**Figure 7 nanomaterials-13-02363-f007:**
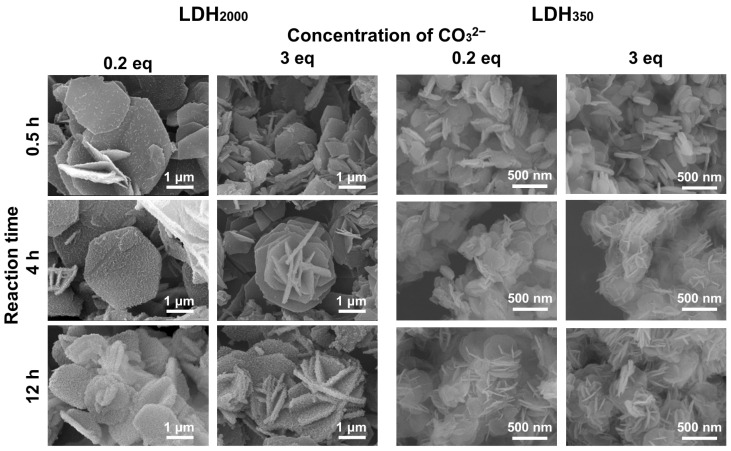
SEM images of LDH_2000_ and LDH_350_ reconstructed with CO_3_^2−^ at the concentrations of 0.2 eq and 3 eq and the reaction time at 0.5 h, 4 h, and 12 h.

**Figure 8 nanomaterials-13-02363-f008:**
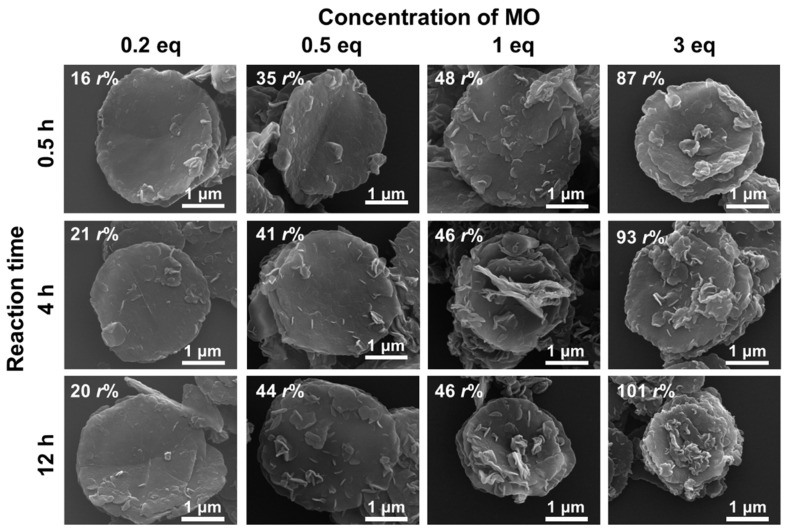
SEM images of LDH_2000_ reconstructed with MO at the concentrations of 0.2 eq, 0.5 eq, 1 eq, and 3 eq and the reaction time at 0.5 h, 4 h, and 12 h with the inset values of the loading ratio, *r*%.

**Figure 9 nanomaterials-13-02363-f009:**
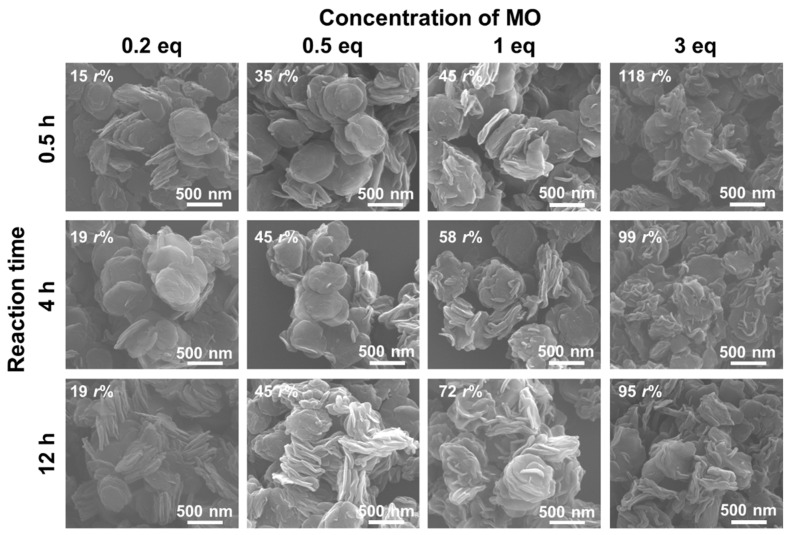
SEM images of LDH_350_ reconstructed with MO at the concentration of 0.2 eq, 0.5 eq, 1 eq, and 3 eq and the reaction time at 0.5 h, 4 h, and 12 h with the inset value of the loading ratio, *r*%.

## Data Availability

The data presented in this study are available upon request from the corresponding author.

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
