# Peer review of "Controlling the Surface Morphology of Two-Dimensional Nano-Materials upon Molecule-Mediated Crystal Growth"

_nanomaterials, 2023, doi:10.3390/nano13162363_

Round 1

Reviewer 1 Report

The authors reports in this manuscript about the controlling the surface morphology of two-dimensional nanomaterials upon molecule-mediated crystal growth. The authors mention that one of the main scope for this work reported here is to find a way for controlling the surface morphology of Mg-Al layered double hydroxide (LDH). However the AFM results, that are missing in the main manuscript, do not clearly indicate that the authours have managed to achieve a good control of the surface morphology of Mg-Al layered double hydroxide. The authors need to revise the manuscript to address this matter and to comment on the AFM results that they have only reported in the supplementary file .

Minor English revision of the manuscript is required. 

Author Response

Comments by Reviewer 1:

  1. The AFM results, that are missing in the main manuscript, do not clearly indicate that the authors have managed to achieve a good control of the surface morphology of Mg-Al layered double hydroxide. The authors need to revise the manuscript to address this matter and to comment on the AFM results that they have only reported in the supplementary file.

Author reply

            We agree with the reviewer’s comment. The AFM results were not sufficient to claim the control of the surface roughness effectively. This is the reason why we chose the SEM images in the main manuscript. We revised the sentence as follows.

p. 11, line 304,

“The 3-dimensional AFM images and height profiles (Figure S2 in Supporting Information) also showed that both loading ratio and reaction time affected the surface roughness Ra of LDH2000, although they were not clear compared to the SEM images.”

p. 12, line 328,

“Moreover, the Ra of LDH350 was governed by both r% values of MO and reaction time as shown in the AFM images in Figure S3 in supporting information (Ra= 0.9 nm at 0.2 eq for 0.5 h and 3 nm at 1 eq for 12 h), although the changes were not sufficiently clear due to small particle size.”

Reviewer 2 Report

The authors propose that the surface morphology of Mg-Al layered double hydroxide (LDH) was successfully controlled by reconstruction during systematic phase transformation from calcined LDH. The higher the concentration of methyl orange, the longer the reaction time, and the rougher the surface obtained at a certain threshold point of methyl orange concentration. The results show that the surface roughness is mediated not only by the guest molecules doped with LDH, but also by the crystal arrangement after containing enough methyl orange. This study is interesting, however, I think that the authors should take into account the following comments:

1. It is important to demonstrate how the surface roughness of the individual LDH particle is controlled through reconstruction. The authors should provide a more detailed explanation of the mechanism of surface roughness of LDH in Figures 1(a) to 1(h).

2. The surface roughness and height profile of all samples were obtained by atomic force microscopy (AFM). Although SEM images were presented, the authors did not provide AFM images and root-mean-square roughness values of the samples as evidence.

Minor editing of English language required.

Author Response

Reviewer 2

  1. It is important to demonstrate how the surface roughness of the individual LDH particle is controlled through reconstruction. The authors should provide a more detailed explanation of the mechanism of surface roughness of LDH in Figures 1(a) to 1(h).

Author reply

            We appreciate the reviewer’s important comment. Although the clear mechanism of reconstruction of LDH is not clear so far, we think the reconstruction was initiated not only the edge of the particles but also flat surface of LDO due to the surface defect. The result of the reconstruction of LDH2000 by CO32− strongly suggested this point. Moreover, we think the roughening mechanism depended on the size, affinity to LDH and flexibility of anions. In order to mention this point, we revised the sentence at

p. 9, line 253,

“The relatively small sized CO32− adsorbed on the surface of LDO2000 and then incorporated through the surface defect with the reconstruction of LDH layer to give rough surface.”

And a sentence was added to

p. 10, line 291,

“It is thought that the rigid LDO structure which is structured by covalent bonds was maintained at less than 45 r% (Figure 1d), while the larger r% than the threshold collapsed the LDO structure to give rough surface (Figure 1g).”

p. 11, line 298,

“Taking into account of the XRD results shown in Figure 5c, a probable reason of the surface roughening at the equilibrium is molecular rearrangement of MO in the interlayer space of the LDH to reach a thermodynamically stable state by π-π interactions [49].”

Moreover, conclusion part was revised as follows,

p. 12, line 337,

“To summarize, the surface roughness of MgAl-LDH was successfully controlled taking advantage of reconstruction of LDO in the presence of small and large guests. The surface roughness was attributed to denaturation of layer stacking by intercalation of small CO32− and large methyl orange molecules into the LDH during the reconstruction of LDO. Change in surface morphology of the LDH by the reconstruction was controlled by the concentration of the small sized guest (CO32−) and the loading ratio of the large sized guest (methyl orange, MO), respectively. An obvious size effect of LDO precursors on resulting morphology after reconstruction by CO32− was observed. The reconstruction of larger sized LDH (2000 nm) at low concentrations of CO32− (0.2 eq) gave the small LDH particles on the external surface to give rough surface (Figure 1e) and sand-rose morphology was given at the high concentration of CO32− (3 eq, Figure 1f), suggesting the possibility that the reaction rate affected the resulting morphology. The reconstruction of LDH with a smaller lateral size (350 nm) by CO32− partially exfoliated LDH layers rather than giving small particles at the surface (Figure 1f). The partial exfoliation of layers was only facilitated in the smaller lateral size in the whole layer region. The surface roughening by intercalation of MO had a threshold of incorporation (~45 %) regardless of the particle size of the pristine LDHs (Figure 1g). The higher incorporation and the longer reaction time gave the rougher surface. As shown in the time-dependent loading ratio values, the MO uptake was equilibrated in early time while the roughening continued afterwards. This strongly suggested that the adsorption and the roughening occurred in different time stage. The CO32−, which has smaller and more rigid structure, and higher affinity to LDH than MO, effectively adsorbed and collapsed the LDO structure to alter the morphology even at low concentrations and in short reaction time; on the other hand, MO, which has more flexible structure than CO32−, could maintain the crystalline structure of LDO until 45 % of leading ratio. It is therefore concluded that by choosing appropriate reaction conditions for reconstruction, such as type of guest molecules, concentration of anions and reaction time, the surface roughness of an individual LDH particle could be controlled. Furthermore, the control of the roughness of the particle surface could expand the application of LDHs and LDOs from versatile nanomaterials to practical application like functional film components.”

  1. The surface roughness and height profile of all samples were obtained by atomic force microscopy (AFM). Although SEM images were presented, the authors did not provide AFM images and root-mean-square roughness values of the samples as evidence.

Author reply

            According to the reviewer’s suggestion, root-mean-square (RMS) values were added in Figure S1-S4 in supporting information. The RMS values in LDH2000 systems included the tilt of the particle, showing large value. However, the trend that RMS increased with longer reaction times. In LDH350 system, the RMS values were hard to be obtained due to the smaller particles which make the edge of the particle unclear. Taking into account of the comment from the reviewer 3 simultaneously, we revised the sentence as follows.

p. 5, line 179,

“As shown in Figure S1 in supporting information, surface roughness Ra was estimated to be 37 nm and 0.8 nm, respectively, for LDH2000 and LDH350 from AFM as shown in Figure S1a and S1c in supporting information. As Ra was and root-mean-square (RMS) values of single LDH particles were less reliable due to tilted arrangement of LDH2000 and unclear particle edge of small particles size of LDH350, we considered these values as a tendency of the surface roughness rather than the quantitative results.”

Reviewer 3 Report

In this manuscript, the authors reported the surface roughening of LDH by reconstruction with carbonate and methyl orange. The as-prepared structures were characterized by XRD, SEM and AFM. The results indicate that the small sized and large sized guest molecules play important role in controlling the surface morphology change of LDH. The work is interesting and the conclusion is well supported by the experimental data. Therefore, I recommend acceptance after the following concerns are well addressed.

1. The measurement of surface roughness, Ra, is not reasonable. In order to compare the true Ra of various LDHs, the whole image should be true surface of LDH. In their work, the individual LDH flake was deposited on substrate with large area. In this case, the thickness shows great influence on the measurement of Ra, not the true surface roughness. This question should be reconsidered carefully to support the conclusion.

2. Why the NaHCO3 was used to investigate the influence of carbonate? Why not use Na2CO3 directly? Does the pH of solution influence the reconstruction? Do other carbonates show similar role?

3. In line 148, page 4, the MOthe should be MOcalc.

4. In line 136, page 4, the scan rate of AFM should be 0.5 Hz, not 0.5 MHz.

5. In line 210, page 7, the Figure 5 shows the result of LDH2000. No result of LDH350 was presented.

Minor editing of English language required.

Author Response

Reviewer 3

  1. The measurement of surface roughness, Ra, is not reasonable. In order to compare the true Ra of various LDHs, the whole image should be true surface of LDH. In their work, the individual LDH flake was deposited on substrate with large area. In this case, the thickness shows great influence on the measurement of Ra, not the true surface roughness. This question should be reconsidered carefully to support the conclusion.

Author reply

            We agree with the reviewer’s comment. The Ra values were hard to obtained due to the tilt of LDH2000 particles on the substrate and unclear edge of LDH350 particles. As SEM image is qualitative data, we thought that it is better to show the Ra values. According to this point and the suggestion from the reviewer 2, we revised the sentences to mention the low reliability of the Ra values as follows.

p. 5, line 179,

“As shown in Figure S1 in supporting information, surface roughness Ra was estimated to be 37 nm and 0.8 nm, respectively, for LDH2000 and LDH350 from AFM as shown in Figure S1a and S1c in supporting information. As Ra and root-mean-square (RMS) values of single LDH particles were less reliable due to tilted arrangement of LDH2000 and unclear particle edge of small particles size of LDH350, we considered these values as a tendency of the surface roughness rather than the quantitative results.”

p. 11, line 304,

“The 3-dimensional AFM images and height profiles (Figure S3 in Supporting Information) also showed that both loading ratio and reaction time affected the surface roughness Ra of LDH2000, although they were rather qualitative compared to the SEM images.”

  1. Why the NaHCO3 was used to investigate the influence of carbonate? Why not use Na2CO3 directly? Does the pH of solution influence the reconstruction? Do other carbonates show similar role?

Author reply

            We think both HCO3 and CO32− has same effect in the reconstruction, as the both anionic species exist majorly in CO32− over pH 9 (current reconstruction condition) due to the acid-base equilibrium (pKa1 and pKa2 of carbonic acid is 6.77 and 9.93 respectively). However, in order to avoid unexpected phenomena by the large amount of cations (Na+ in this case), we would like to fix the concentration of Na cations in CO32− and MO systems. Thus, we finally chose NaHCO3. The results in this paper suggested that the reactions with CO32− which has a special preference to LDH and with larger MO decomposed the original particle morphology effectively, meaning that the slow reaction with smaller anions may return from LDO to original LDH which is known as memory effect. Thus, we think the type of the anion including size, affinity to LDH and molecular flexibility is important rather than counter anion. In order to mention this point, the following part was revised.

p. 4, line 113: NaHCO3 was used in the reconstruction of LDO2000 and LDO350 as a carbonate source and the amount of Na+ concentration was set same with that for MO.

  1. In line 148, page 4, the MOthe should be MOcalc.

Author reply

            We appreciate the reviewer’s careful checking. The part was revised as follows,

p. 4, line 155: (1)

  1. In line 136, page 4, the scan rate of AFM should be 0.5 Hz, not 0.5 MHz.

Author reply

            We appreciate the reviewer’s careful checking. We revised the sentence as follows.

p. 4, line 142: “The sample images and height profiles were obtained by non-contact mode with 0.5 MHz scan speed.”

  1. In line 210, page 7, the Figure 5 shows the result of LDH2000. No result of LDH350 was presented.

Author reply

            We appreciate the reviewer’s suggestion. We added the X-ray diffraction patterns of reconstructed LDH350 in Figure 4. The XRD peaks of LDH increased and those of LDO decreased more clearly than those in LDH2000 systems. Moreover, the particle sizes estimated from the Sherrer’s equation were larger than those obtained from LDH2000 systems, which supported the SEM observations. We added this point as follows.

p. 6, line 213,

“The clearer growth of the peaks attributed to LDH and decreasing of peaks of LDO were shown in LDH350 systems compared to LDH2000 systems (Figure 4c and 4d). The higher surface area originated to the smaller particle size of LDH350 occurred effective adsorption and reconstruction of LDH layers. As summarized in Table S3 in supporting information, the particle sizes of the reconstructed LDH for 0.5 h reaction estimated by Sherrer’s equation were 8.02 nm at 0.2 eq and 9.03 nm at 3 eq, which were comparable to the ones obtained in the LDH2000 system for 12 h reactions. It suggested that the quick hydration along ab-plane reconstructed the larger area simultaneously to give the larger LDH particles.”

p. 7, line 223: Figure 4 and the figure caption were revised as follows,

Figure 4. Powder XRD patterns of (a and b) reconstructed LDH2000 and (c and d) LDH350, where bottom black lines: pristine LDHs (LDH2000 and LDH350), top black lines: LDO precursors (LDO2000 and LDO350) and reconstructed LDHs with CO32− at the reaction time of blue lines: 0.5 h, green lines: 4 h and red lines: 12 h at the concentrations of CO32− of (a and c) 0.2 eq and (d and d) 3 eq.

Reviewer 4 Report

Authors of the paper reported a method that can control the reconstruction process from Mg-Al layered double oxide (LDO) to Mg-Al layered double hydroxide (LDH) to achieve different LDH particle roughness. I think the argument in the paper about the relationship between competing crystal growth and surface roughness is interesting. Therefore, I recommend this work can be acceptable for publication in Nanomaterials after several questions are addressed.

1. The position of the supporting information mentioned in the text needs to be recalibrated. For example, the crystalline size of the reconstructed LDH in row 197 corresponds to Table S2 instead of Table S1.

2. As seen in Table S2 of the supporting information, the crystalline size of the reconstructed LDH after 12h at 3eq was smaller than that after 4h at 3eq. Please explain it.

3. The picture symbol format and font size in figure 2, 4, 5, and 6 should be unified.

4. Why is there no description and interpretive language for figure 6?

5. Where are SEM images of LDH350 reconstructed with CO32- at the concentrations of 0.2eq and 3eq and reaction time at 0.5h, 4h and 12h?

6. What is the difference between CO32- and organic methyl orange (MO) anions on the growth of crystals in the ab-plane and on the stacking along c-axis, and what is the relationship between the competitive crystal growth statement mentioned in the article and the effect of these two anions on crystal growth.

Author Response

Reviewer 4

  1. The position of the supporting information mentioned in the text needs to be recalibrated. For example, the crystalline size of the reconstructed LDH in row 197 corresponds to Table S2 instead of Table S1.

Author reply

            We appreciate the reviewer’s careful checking. We revised the Table numbers as follows.

p. 6, line 207; As summarized in Table S2 in supporting information,…

Moreover, according to the addition of Figure S2 in supporting information, the sentence was revised as follows.

p. 11, line 304: The 3-dimensional AFM images and height profiles (Figure S3 in Supporting Information) also showed that…

p. 12, line 328: Moreover, the Ra of LDH350 was governed by both r% values of MO and reaction time as shown in the AFM images in Figure S4 in supporting information (Ra= 0.9 nm at 0.2 eq for 0.5 h and 3 nm at 1 eq for 12 h),…

  1. As seen in Table S2 of the supporting information, the crystalline size of the reconstructed LDH after 12h at 3eq was smaller than that after 4h at 3eq. Please explain it.

Author reply

            We think it is one of the important points of this study. As shown in Figure 7 in the revised manuscript, the reconstruction of LDH2000 at 3 eq for 4 h gave rough surface by giving sand-rose structure. The particle became to be covered with small debris at reaction time 12 h. We think the difference of the morphological change was originated in the adsorption rate (or concentration difference) of CO32−. The decrease of the crystallite size from 4 h to 12 h in Table S2 was thought to reflect this morphological change: i) until 4 h: fast adsorption of CO32− at early time stage develops the general sand-rose structure and ii) between 4 h and 12 h: slow adsorption of CO32− at later time stage modifies the morphology in small scale, e.g. formation of small particles on surfaces. The effect of adsorption rate on the generated particle size was supported by the reconstruction at 0.2 eq, where small particles appeared due to slow adsorption. In order to explain this point, the following part was revised.

p. 9, line 263,

“The massive consumption of CO32− before 4 h reaction made the overall sand-rose morphology; and the slow adsorption of CO32- after 4 h reaction induced rough surface by generating small particles on the sand-rose surface. The phenomena was corresponded to the decrease of the particle size estimated from Sherrer’s equation from 19.3 nm to 9.01 nm as summarized in Table S2 in supporting information.”

  1. The picture symbol format and font size in figure 2, 4, 5, and 6 should be unified.

Author reply

            We appreciate the reviewer’s careful checking. The following parts were revised.

Figure 2. Powder XRD patterns of (a) LDH2000, (b) LDH350 divided 10, (c) LDO2000 and (d) LDO350 multiplied 5, respectively.

Figure 3. Scanning electron microscopic (SEM) images of (a) LDH2000, (b) LDO2000, (c) LDH350 and (d) LDO350, respectively.

Figure 4. Powder XRD patterns of (a and b) reconstructed LDH2000 and (c and d) LDH350, where bottom black lines: pristine LDHs (LDH2000 and LDH350), top black lines: LDO precursors (LDO2000 and LDO350) and reconstructed LDHs with CO32− at the reaction time of blue lines: 0.5 h, green lines: 4 h and red lines: 12 h at the concentrations of CO32− of (a and c) 0.2 eq and (b and d) 3 eq.

Figure 5. Powder XRD patterns of (black line) LDH2000 and reconstructed LDHs at the reaction time of 0.5 h (blue line), 4 h (green line) and 12 h (red line) at the concentrations of MO of (a) 0.2 eq, (b) 0.5 eq, (c) 1 eq and (d) 3 eq.

Figure 6. Powder XRD patterns of (black line) LDH350 and reconstructed LDHs at the reaction time of 0.5 h (blue line), 4 h (green line) and 12 h (red line) at the concentrations of MO of (a) 0.2 eq, (b) 0.5 eq, (c) 1 eq and (d) 3 eq.

p. 3, line 79; Figure 1. Schematic mechanism of surface roughening of LDH by systematic reconstruction with CO32− and MO.

p. 10, line 278; Figure 7. SEM images of LDH2000 and LDH350 reconstructed with CO32− at the concentrations of 0.2 eq and 3 eq and reaction time at 0.5 h, 4 h and 12 h.

  1. Why is there no description and interpretive language for figure 6?

Author reply

            We appreciate the reviewer’s careful checking. The following part was revised.

p. 8, line 232; Figure 5 and 6 represent low angle powder XRD patterns of LDH2000 and LDH350 before and after reconstruction with MO at various concentrations and reaction times.

  1. Where are SEM images of LDH350 reconstructed with CO32- at the concentrations of 0.2eq and 3eq and reaction time at 0.5h, 4h and 12h?

Author reply

            We appreciate the reviewer’s suggestion. According to the reviewer’s kind suggestion, we carried out additional experiments on the CO32- reconstructed LDH and measured XRD and SEM. The XRD patterns and SEM images in Figure 4 and 7, were newly added, respectively. Briefly, in LDH2000 system, adsorption rate dependence on the morphological change was observed, while change in morphology of LDH350 was not depended on the concentration of CO32− due to its smaller particle size. The reconstruction LDH350 seemed to be accompanied with the exfoliation of LDH layers to give sand-rose structure and thinner layer. It is clear size effect on the change in morphology by the reconstruction. We mentioned this point in revised manuscript as follows.

Abstract (p. 1, line 16)

“During the reaction, the degree of crystal growth along ab-plane and stacking along c-axis was significantly influenced by the molecular size and the reaction conditions but not dependent on the particle size of the pristine LDH. The lower concentration of carbonate gave smaller particles on the surface of larger LDO (2000 nm), while the higher concentration induced a sand-rose structure. The reconstruction of smaller sized LDH (350 nm) did not depend on the concentration of carbonate due to effective adsorption and gave sand-rose structure and exfoliated LDH layers.”

p. 10, line 267,

“The reconstruction for 0.5 h with 0.2 eq and 3 eq of CO32− did not altered the morphology of LDO350 significantly (Figure 7 right columns). The reconstruction at 0.2 eq for 4 h exfoliated a part of the LDH layers in sand-rose like structure and thus thinner LDH particles were obtained by 12 h reaction. The similar change of the morphology was observed by the reaction at 3 eq for 4 h. The 12 h reaction time gave clear sand-rose and thinner particles. It is worth noting that the small particles which were observed in the reconstruction of LDH2000 were not observed in the LDH350 systems, suggesting the size effect on the reconstruction by CO32−. The larger lateral size of LDH2000 would not allow the exfoliation of whole one layer by the reconstruction, while the whole one layer of LDH350 was exfoliated to give the thinner particles.”

According to the above discussion, Figure 1 was revised as follows.

Figure 1. Schematic mechanism of surface roughening of LDH by systematic reconstruction with CO32− and MO.

  1. What is the difference between CO32- and organic methyl orange (MO) anions on the growth of crystals in the ab-plane and on the stacking along c-axis, and what is the relationship between the competitive crystal growth statement mentioned in the article and the effect of these two anions on crystal growth.

Author reply

            We appreciate the reviewer’s important comment. The parameter to give rough surface in the LDH through reconstruction differs according to the anion type. We hypothesized that the molecular size, affinity to LDH and flexibility of anion affect the surface morphology of LDH upon reconstruction. Carbonate (CO32−) is one of the most small anion to be intercalated in the LDH, has extreme affinity toward LDH layer due to its planar structure and divalence, and rigidity. Thus, CO32− adsorbed on LDO to change the morphology effectively. At high concentration, which corresponds to the faster adsorption of anions, the hydration along to the ab-plane occurred quickly to give sand-rose structure and partially exfoliated layer; on the other hand, at low concentration, the hydration occurred slowly to give small particles at the surface.

On the other hand, MO is fairly large and long molecules with moderate affinity to LDH and flexibility. As the affinity of MO toward LDH is less than CO32−, we expected the less effective adsorption of MO on LDO. The flexible structure of MO compared with CO32− facilitated the hydration of LDO along ab-plane without seriously disturbing the original structure along c-axis. Thus, it was thought that the MO system had threshold to control the surface roughness. In order to mention this point, the part was added as follows.

p. 9, line 258,

“It is thought that the adsorption of CO32− was slow at the low concentration of 0.2 eq, so that the surface along to ab-plane was hydrated to give small particles of metal hydroxide about 5 nm. On the other hand, the adsorption of CO32− was fast at high concentration of 3 eq to develop CO32- intercalated particles and to collapse larger ab-plane – formation of relatively large particles about 20 nm – resulting in that the sand-rose structure at 4 h. The massive consumption of CO32− before 4 h reaction made the overall sand-rose morphology; and the slow adsorption of CO32- after 4 h reaction induced rough surface by generating small particles on the sand-rose surface. The phenomena was corresponded to the decrease of the particle size estimated from Sherrer’s equation from 19.3 nm to 9.01 nm as summarized in Table S2 in supporting information.”

Moreover, the part was revised as follows.

p. 10, line 291,

“It is thought that the rigid LDO structure which is structured by covalent bonds did not accept the increase of c-axis (interlayer space) induced by crystal growth of LDH along to ab-plane at less than 45 r%, while the larger r% than the threshold collapsed the LDO structure to give rough surface.”
p. 11, line 298,

“Taking into account of the XRD results shown in Figure 5c, a probable reason of the surface roughening at the equilibrium is molecular rearrangement of MO in the interlayer space of the LDH to reach a thermodynamically stable state by π-π interactions [49]. It is worthy to note here that the small particles which were observed in the CO32− systems were not observed even at high loading ratio (~100 r%), which suggests that the size of intercalated molecules affected the morphology of the reconstructed LDHs.”

p. 13, line 357,

“The CO32−, which has a smaller and more rigid structure, and higher affinity to LDH than MO, effectively adsorbed and collapsed the LDO structure to alter the morphology even at low concentration and short reaction time; on the other hand, MO, which has more flexible structure than CO32−, could maintain the crystalline structure of LDO until 45 % of leading ratio.”

Round 2

Reviewer 2 Report

The authors have revised and responded to the comments.